# Burnout Impact of COVID-19 Pandemic on Health-Care Professionals at Assiut University Hospitals, 2020

**DOI:** 10.3390/ijerph18105368

**Published:** 2021-05-18

**Authors:** Shimaa A. Elghazally, Atef F. Alkarn, Hussein Elkhayat, Ahmed K. Ibrahim, Mariam Roshdy Elkhayat

**Affiliations:** 1Occupational and Environmental Medicine Department, Assiut University, Asyut 715715, Egypt; shima_dola@aun.edu.eg; 2Chest Disease Department, Faculty of Medicine, Assiut University, Asyut 715715, Egypt; Afaroukeg@yahoo.com; 3Cardiothoracic Surgery Department, Faculty of Medicine, Assiut University, Asyut 715715, Egypt; elkhayat@aun.edu.eg; 4Public Health Department, Faculty of Medicine, Assiut University, Asyut 715715, Egypt; ahmed.ibrahim@aun.edu.eg

**Keywords:** burnout syndrome, Maslach Burnout Inventory, COVID-19

## Abstract

Background: burnout syndrome is a serious and growing problem among medical staff. Its adverse outcomes not only affect health-care providers’ health, but also extend to their patients, resulting in bad-quality care. The COVID-19 pandemic puts frontline health-care providers at greater risk of psychological stress and burnout syndrome. Objectives: this study aimed to identify the levels of burnout among health-care professionals currently working at Assiut University hospitals during the COVID-19 pandemic. Methods: the current study adopted an online cross-sectional design using the SurveyMonkey^®^ website for data collection. A total of 201 physicians were included and the Maslach Burnout Inventory (MBI) scale was used to assess the three burnout syndrome dimensions: emotional exhaustion, depersonalization, and reduced personal accomplishment. Results: about one-third, two-thirds, and one-quarter of the respondents had high emotional exhaustion, high depersonalization, and low personal accomplishment, respectively. Younger, resident, and single physicians reported higher burnout scores. The personal accomplishment score was significantly higher among males. Those working more than eight hours/day and dealing with COVID-19 patients had significantly higher scores. Conclusion: during the COVID-19 pandemic, a high prevalence of burnout was recorded among physicians. Age, job title, working duration, and working hours/day were significant predictors for burnout syndrome subscale results. Preventive and interventive programs should be applied in health-care organizations during pandemics.

## 1. Introduction

Burnout syndrome (BOS) is a common occupational and public-health problem, and recently, its importance and rates have been rising [1]. It is defined as a state of psychological, emotional, and physical stress that occurs due to prolonged periods of exposure to chronic occupational stressors [2]. Moreover, workers in highly stressful jobs are at higher risk of job burnout, such as health-care workers (HCWs) during the COVID-19 pandemic. As HCWs have to take care of infected patients, they face higher rates of infection/fatality and the fear of transmitting the infection to their families. All of these factors lead to high social and mental pressure on HCWs dealing with the current pandemic [3,4]. BOS has three dimensions: feelings of emotional exhaustion (EE) (depletion of emotional resources), depersonalization (DP) (developing cynical attitudes about patients), and a sense of low personal accomplishment (PA) [5]. BOS was first described by Maslach et al., in 1996 [6], and was commonly observed in those who work with people and whose jobs are characterized by high levels of work-related stress. It may result from a combination of individual risk factors and organizational stressors [7].

Moreover, working in a stressful environment may affect the emotional stability of medical staff [8]. Thus, BOS was recognized as a serious problem which is common and increasing among medical staff, especially those dealing with critically ill patients, i.e., they work in a stressful environment where they have to take care of their patients, and react physically and emotionally to each patients’ problem [9]. According to the Critical Care Societies Collaborative (CCSC), up to 45% of critical-care physicians reported symptoms of severe BOS [10]. Previous studies have reported high prevalence of burnout among health-care providers; ranging from 25 to 75% worldwide [8,11]. In the Arab countries, research showed that rates of BOS were higher among females, young, unsatisfied physicians, those not exercising, and those who had fewer years of experience [12,13,14]. In Egypt, a cross-sectional study of 168 health-care providers was carried out in Zagazig University hospitals to assess the prevalence of BOS, which found high rates of EE (53%), DP (64.3%), and PA (67.3%) among the studied physicians [15].

BOS has several adverse outcomes: it affects not only the psychological and physical health of HCWs, through depression, insomnia, and gastrointestinal problems, but also the patient, in the form of bad-quality care provided by the affected staff and low patient-satisfaction. The negative outcomes of BOS on HCWs extend to affect the health-care organization as it can result in higher absenteeism, poor job satisfaction, repeated turnover, low morale in the staff, and financial losses. It is important to identify factors that contribute to burnout, and to develop strategies for preventing and treating burnout among health-care providers, in order to create and sustain a healthy and productive workplace [16].

The COVID-19 pandemic is an ongoing pandemic of coronavirus disease 2019 (COVID-19) which is caused by severe acute respiratory syndrome coronavirus 2 (SARS-CoV-2). It is a highly infectious disease with a long incubation period. It was first identified in December 2019 in Wuhan, China. The World Health Organization (WHO) declared the outbreak a public health emergency of international concern in January 2020, and a pandemic in March 2020 [17]. As of 4th of April 2021, about 130.25 million cases had been confirmed, with 2.8 million deaths attributed to COVID-19 reported globally by WHO [18]. The COVID-19 pandemic puts frontline HCWs at great risk of psychological stress. The prevalence of high-level stress, depressive symptoms requiring treatment, and anxiety symptoms requiring further evaluation among HCWs were 3.7%, 11.4%, and 17.7%, respectively [19]. Approximately 10% of confirmed COVID-19 cases involved health-care providers [20].

Additionally, high levels of burnout in HCWs were reported during this pandemic. Many factors were identified as contributing to the psychological impact of this pandemic on health-care providers, such as the fatal nature of COVID-19, the lack of effective treatment, the higher rate of infection and mortality among them, being away from their families, and social stigmatization. Additionally, many organizational factors contributed to burnout among HCWs during this pandemic, such as longer working hours, shortage of Personal Protective Equipment (PPE), heavy workload, extensive responsibilities, lack of specific drugs, protocols and care for unstable patients [19,21], presence of chronic disease, and continuous changes to the locations and tasks of doctors [22,23].

BOS was represented as an increasingly serious problem among medical staff especially during the COVID-19 pandemic and was associated with difficult working conditions and feelings of dissatisfaction at work [4]. Most studies on BOS among physicians during the COVID-19 pandemic were conducted in developed countries, which had fewer numbers of infected patients, than in developing ones such as Egypt [24,25,26]. The current study aimed to assess the levels of BOS and its subscales among physicians currently working at Assiut University hospitals and to identify the most important correlates of BOS.

## 2. Materials and Methods

An online cross-sectional study was conducted via the SurveyMonkey^®^ website [27] during the current COVID-19 pandemic to evaluate the levels of burnout among physicians working at Assiut University hospitals. The study was conducted in the period between June 2020 and July 2020

Sample-size calculation was carried out using an EPI info 2000 statistical package [28]. The calculation was based on an expected frequency of burnout in physicians dealing with COVID-19 cases of 13% [29] during the COVID attack with a difference of 5%, and a confidence interval of 95%. The minimum sample required was 174 doctors. Nearly 600 physicians were invited to participated in the study, the response rate was 35% (201 doctor completed questionnaire). A convenience-sampling technique was applied in this study.



### 2.1. Ethical Consideration

The protocol was reviewed by Assiut University ethical committee. and was recorded on the clinical-trial registration website (No. NCT04363229). All participants were asked to sign an informed-consent document, which was clearly stated and indicated the purpose, procedures, pros, and cons of the study. Furthermore, confidentiality and anonymity were assured with no incentives or rewards for the participants. The study followed Declaration of Helsinki guidelines [30].

### 2.2. Data Collection Tools

A predesigned self-fulfillment questionnaire was prepared as an online version for the assessment of the levels of burnout and its determinants among the studied population.

The questionnaire was composed of three parts:(a)Socio-demographic data of the studied population, such as age, residence, marital status, smoking history, and history of chronic diseases.(b)Work-related characteristics, such as job title, enrolled department, working system, working hours per day, working days per week, and history of dealing with COVID-19 patients.(c)Maslach Burnout Inventory (MBI) which is universally accepted as the gold-standard self-reported measure due to its high reliability and validity. As the Cronbach’s α value for the main MBI scale was 0.829, the emotional exhaustion subscale was 0.887, the depersonalization subscale was 0.768, and the diminished personal accomplishment subscale was 0.891 [31]. It was designed to assess the three components of the BOS: emotional exhaustion (EE), depersonalization (DP), and reduced personal accomplishment (PA). It is a 22-item questionnaire on a seven-point Likert-scale (ranging from 0 = never to 6 = every day). Scores for each section were obtained by adding the numeric responses of the items which corresponded to each scale. High scores for the first two dimensions (EE and DP) and low scores for the third dimension (PA) indicated BOS [6]. MBI scores were further used to classify participants as having low (≤17 points), moderate (18–29 points), and high (≥30 points) levels of EE, low (< 6 points), moderate (6–11 points), and high (≥12 points) levels of DP and For PA low (≥40 points), moderate (39–34 points), and high (≤33 points) levels burnout dimensions.

### 2.3. Procedure

Several literary works were reviewed, the questionnaire was revised by all authors, and the MBI score was translated into Arabic by specialized translators. The link was then launched on the SurveyMonkey^®^ website and was sent to all of the physicians at Assiut University via their official e-mails. This was followed by repeated reminders via e-mail, phone call, or the WhatsApp application.

## 3. Results

### 3.1. Sociodemographic Characteristics of Enrolled Doctors

Table 1 describes the sociodemographic data of the sampled doctors. It was found that about two-thirds were female and that 90% of the sample lived in urban areas. Moreover, 44.3% of the enrolled doctors were aged 20–30 years, 36.3% were aged 30–40 years, and 19.4% were aged more than 40 years. Noticeably, only 7% were smokers.

### 3.2. Figure 1: The Distribution of the Studied Cohort According to Maslach Burnout Inventory (MBI) Categories

The overall prevalence of BOS was 6% with a mean of 60.6 ± 26 points. For the subscales, high EE, DP and PA scores were observed in 35.5%, 70.6% and 26.4% of the studied sample, respectively in Figure 1.

### 3.3. Table 2: The Association between MBI Sub-Scores and Sociodemographic Characters of Cohort Doctors

Regarding the relationship between MBI scores and sociodemographic characters: significantly (*p* = 0.01) higher PA scores were reported by males (32.7) compared to females (28.9). Contrarily, the results of the current study did not detect any statistical difference between males and females regarding the mean EE and DP. Furthermore, a downward linear trend was found in the mean EE and DP for age; the youngest age group (20–30 years) recorded the highest mean EE and DP (28.3 and 22.9, respectively), and the oldest age group (>40 years) recorded the lowest mean EE and DP (17.5 and 1.6, respectively). Conversely, the youngest group (20–30 years) recorded the lowest mean PA (27.1) and the oldest age group (>40 years) recorded the highest mean PA (36.7). These associations were statistically significant (*p* < 0.001) (Table 2). Moreover, there was a statistically significant relationship between the marital status of enrolled physicians and their MBI subscale scores: single physicians recorded higher EE (26.6) and DP (22.8) mean scores and lower PA mean scores (27.1) than married doctors (*p* < 0.05). On the other hand, other factors such as residence, smoking, and chronic disease history were not statistically associated with their MBI subscale scores (Table 2).

### 3.4. Table 3: The Association between MBI Sub Scores with Their Workplace Characteristics and History of COVID-19 Cases Exposure

Concerning job title, resident doctors showed the highest scores for EE and DP, and lowest PA scores, followed by assistant lectures, then specialist/Lecturer, and finally, Assistant prof/professors (*p* < 0.05). Oppositely, their EE and DP mean scores were not affected by their medical subspecialty, but PA scores were recorded to be significantly higher among doctors in Surgical/Anesthesia and ICU departments (*p* < 0.001) (Table 3). Although, there were significantly higher DP scores among physicians with mixed shifts (19.9) than among others working only morning shifts (16.5), their EE and DP mean scores were not affected by this working system. Moreover, physicians attending work for more than 8 h per day recorded the highest EE and DP mean scores compared to workers working 4–8 h and <4 h (*p* < 0.001). Regarding histories of dealing with COVID-19 patients: statistically, only the DP mean score was found to be significantly higher among physicians who had a history of dealing with COVID-19 patients (21.6) when compared to physicians who did not have a history of dealing with COVID-19 (17.1) (*p* = 0.01). Although the EE score was recorded to be higher in physicians who had treated COVID-19 patients historically than those who had not, this did not show a statistically significant difference, while both had the same PA score (Table 3).

### 3.5. Figure 2 Correlations between Working Duration and Working Days per Week with Maslach Burnout Inventory Subfield Scores

Multicollinearity diagnosis was depicted in Figure 2. This shows a moderately significant negative correlation of EE and DP scores of enrolled doctors with regards to their working experience (r = −0.33, *p* ≤ 0.001 and r = −0.39, *p* ≤ 0.001, respectively). Oppositely, a significant positive mild correlation was found with their working days per week. Moreover, PA scores of all of the doctors recorded significant mild positive correlation with working duration only (r = 0.28 and *p* ≤ 0.001).

## 4. Discussion

During the COVID-19 pandemic, several studies revealed a high prevalence of stress and burnout among health-care workers [32], thus affecting their quality of life as well as the quality of health service provided [33]. Assiut University hospital is one of the largest tertiary-level hospitals in Egypt.

The current study revealed that about one-third (35.5%), two-thirds (70.6%), and one-quarter (26.5%) of the respondents recorded high EE and DP, and low PA, respectively. Also, 6% had BOS (high scores for the first two dimensions (EE and DP) and low scores for the third dimension (PA)). This was consistent with a multinational study carried out in 45 countries to explore the prevalence of BOS among health-care professionals during the first wave of the COVID-19 pandemic. It was found that about half (56%) of the cohort showed high EE, 48.9% recorded high depersonalization, and about one-third (38%) showed low PA [34].

Across the world, burnout prevalence was lower than that reported in this research: Australia (30%) [35], Brazil (21%) [36], Wuhan (FL 13% vs. UW 39%) [29], Italy (37%, 25%, 15.3%) [37] and Spain (41%, 15.2%, 8.4%) [33]. Unlikely, the prevalence was higher in Portuguese HCWs (53%) [38,39]. This diversity of results could be attributed to the different scales used, cultural differences, and dissimilar health systems.

Concerning gender effect on BOS: statistically, PA scores were significantly higher among males than females (*p* = 0.01), while EE and PD scores were not affected by gender. This matched with a study in New Zealand [39]. Other studies recorded no difference [31], thus indicating that females were more liable to burnout than males. This could be explained by their loads of household work which, in addition to their career responsibilities, could raise stress [40]. On the other hand, other studies reported that more females were liable to experience EE [41,42].

In the current work, age was found to be a significant factor affecting MBI score, as the youngest age group (20 < 30) recorded the highest EE and DP, and the lowest PA scores (*p* < 0.001). In agreement with these findings, an Indian study was carried out during the COVID-19 pandemic among HCWs [43], finding that burnout was higher among the younger population. This finding was supported by previous studies conducted before the pandemic attack in different areas [44,45,46]. This could be attributed to the workload and lack of working experience at the beginning of their careers; additionally, juniors have more contact on the frontline with COVID-19 patients than senior HCWs [47]. On the other hand, Salem et al., 2018, were not consistent with our results as they found that there was no significant relationship between age and burnout, and reported that increasing age significantly increased the perception of EE (β = 0.247, *p* = 0.019), while it had no significant impact on DP or PA (*p* > 0.05) [48].

Concerning marital condition, single doctors were more liable to burn out than married doctors as they had significantly higher EE and DP scores and lower PA scores. Our results matched with a multicentric Egyptian study on medical oncology professionals which reported that being married was significantly associated with higher levels of personal accomplishment; this may be due to the support given by a partner. Conversely, single women can be put under great work and social stress by our Middle Eastern society [49]. The current results were comparable to those of the New Zealand national survey in which being single was associated with higher levels of EE [39]. Likewise, it was in accordance with a cross-sectional study among HCWs in Aswan University Hospital [31]. On the other hand, studies conducted in Saudi Arabia [45] and Egypt [50] concluded that higher BOS levels were strongly associated with married respondents.

The current study revealed that resident doctors had the highest scores for EE and DP, and the lowest PA scores. This was supported by the moderate negative correlation found between working experience years and both EE and DP scores of the enrolled doctors, and by the mild positive correlation between PA score and number of working experience years. These findings were consistent with a study conducted in Pakistan which reported higher DP scores in doctors with less than 10 years’ experience and higher PA scores among older doctors [51]. In 2013, a study of physicians in a tertiary hospital in Saudi Arabia concluded that the physicians who were most affected by burnout syndrome were resident physicians [45,52].

Furthermore, a systematic review of studies on BOS prevalence among health-care professionals (HCP) in the Arab countries, according to the Preferred Reporting Items for Systematic reviews and Meta-Analyses (PRISMA) guidelines [53] reported high levels of burnout, with the highest prevalence in the EE and DP and lowest PA among resident doctors. This may be explained by the high workload placed upon them, long working hours, and low salary. Additionally, experienced senior doctors become more skilled and committed to their work; therefore, they may be calmer and more capable of facing and managing occupational stressors. They may feel more successful in their profession, thus the lower levels of, and greater adaption to, burnout among them [54,55]. Compared to other Egyptian studies, no significant association between years’ of work and burnout occurrence was reported by Abdallah, 2019 [15], and a significant positive correlation between burnout subscales and years’ of work were reported by Abd EL Latief [56]. The medical department staff were the first line of defense against COVID-19 and academic department staff were overloaded with online learning. However, most unnecessary surgeries were postponed during the attack, thus keeping the workloads of surgical staff lower during the attack than those of others. Statistically, doctors with mixed shifts had significantly higher DP scores than doctors with only morning shifts. This was consistent with a French study [57] which reported that increased frequency of night shifts/month were associated with increased risk of burnout. Our findings also matched the results of a systematic review of studies on BOS prevalence among Arab HCPs that concluded that DP was higher among nurses who worked night shifts and rotating shifts [58]. Moreover, a Lebanese study [59,60] reported that DP scores were higher among nurses who work night shifts and rotating shifts compared to daytime workers. This study revealed that working more than 8 h per day was associated with higher EE and DP scores than working 4–8 h and <4 h. The finding of the current research was similar to two studies conducted in Yemen, 2009 [59], and Lebanon, 2010 [46], which reported a significant association between burnout and long working hours per week. Another study, conducted upon USA military treatment facilities, reported that working more hours was independently predictive of higher burnout scores [41].

Additionally, doctors dealing with COVID-19 patients had higher EE and DP scores, but the difference was only statistically significant for DP as this was one of four significant predictors for DP (dealing with COVID-19 patients increased the DP scores by three points). The current findings matched a study conducted in China during the COVID-19 pandemic that found high prevalence of burnout among frontline nurses [60]. Prior to this pandemic, HCWs were already exposed to burnout risk. This pandemic has exacerbated existing risks and triggered new risks as it led to a sharp increase in admissions of critically ill COVID-19 patients to hospitals, which led to an increase in the workload of HCWs. Additioally, HCWs faced risk of exposure to the COVID-19 infection, long working hours, critical decision making, fatigue, and the fear of spreading the infection to family members. All of this leads to more psychological distress and higher levels of burnout faced by frontline health-care providers [38].

Strengths and Limitations:

This study has encountered several limitations: firstly, the cross-sectional nature lacks causal effect and might jeopardize its external validity; secondly, the online survey method of data collection, which limits the sample to social media-active people (resulting in a low representation of those aged >40 years, whose response was less than younger respondents) might have led to selection bias; thirdly, sample selection was obtained via a convenience-based non-probability technique which may result in lack of representation of all medical classes and limit its generalizability.

## 5. Conclusions

The findings of this study offered an early insight into the serious problems facing physicians who currently work at Assiut University hospitals, especially during the COVID-19 pandemic, which might affect them in negative way. This study provides some guidance for possible interventions. About one-third, two-thirds, and one-quarter of the respondents had high EE, DP, and low PA, respectively. Younger, resident, and single physicians reported higher burnout scores. Personal accomplishment scores were significantly higher among males. Those working more than eight hours/day and those dealing with COVID-19 patients had significantly higher scores. Likewise, age, job title, working duration, and working hours/day were significant predictors for BOS subscale. Preventive and interventive programs should be applied in health-care organizations during pandemics.

## 6. Recommendations

Burn out is one of the most important problems facing health-care physicians, especially during pandemic status (i.e., COVID-19). Hence, all hospitals should prioritize the planning and implementations of strategies to face the situation and support doctors, such as psychological clinics and hotline systems for HCWs, especially physicians. Improvements to work ergonomics and environment (e.g., provide more physicians on each shift, if possible, to decrease workload and assist the occupational health clinic in protecting HCWs) should be enforced by legislations.

## Figures and Tables

**Figure 1 ijerph-18-05368-f001:**
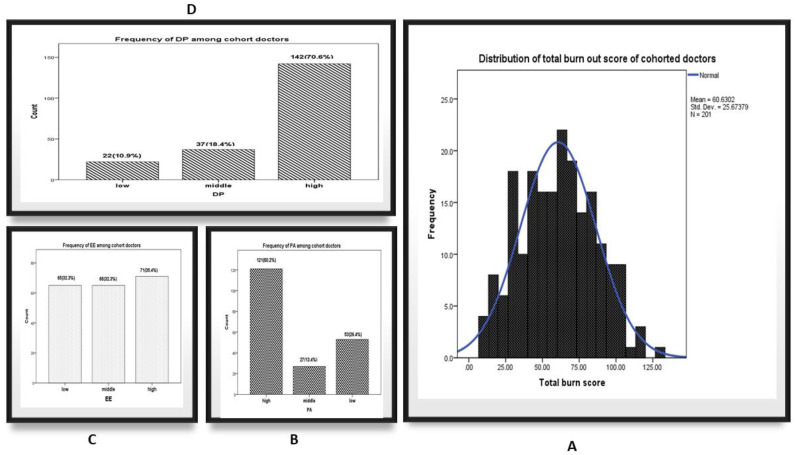
(**A**): Histogram graph of doctors Total MBI score, (**B**): Bar Charts of a sense of personal accomplishment (PA) Categories of enrolled doctors, (**C**) Bar Charts of emotional exhaustion (EE) Categories of enrolled doctors, (**D**) Bar Charts of depersonalization (DP) Categories of enrolled doctors.

**Figure 2 ijerph-18-05368-f002:**
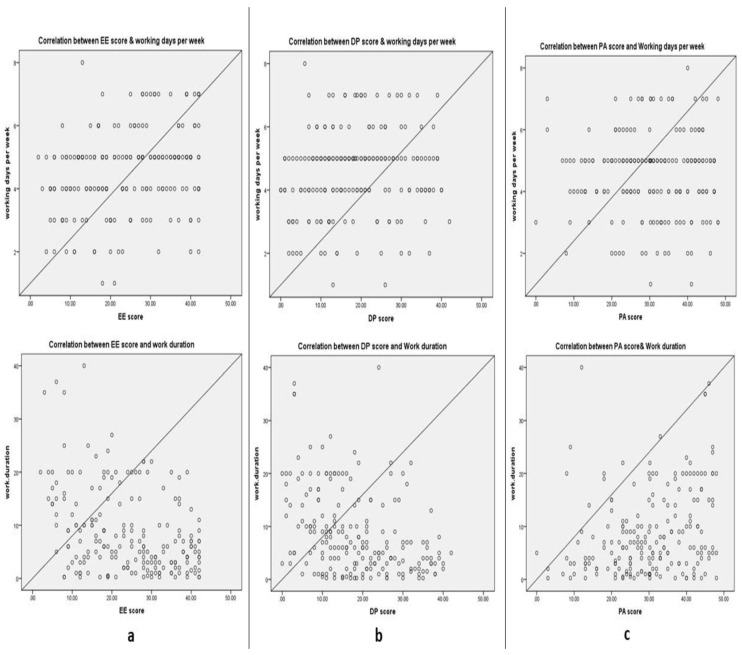
Correlation between working duration and working days per week of physicians and their EE score (**a**), DP score (**b**), and PA score (**c**).

**Table 1 ijerph-18-05368-t001:** Sociodemographic Characteristics of cohort Doctors.

Characteristics	Frequency	Percentage
Gender		
Male/female	70/131	34.8%/65.2%
Age group		
20-/30-/40-	89/73/39	44.3%/36.3%/19.4%
Residence		
Urban/rural	180/21	89.6%/10.4%
Marital Status		
Single/Married	78/123	38.8%/61.2%
Smoking History		
Non-smokers/Smokers	187/14	93%/7%

**Table 2 ijerph-18-05368-t002:** Distribution of doctors Maslach Burnout Inventory mean scores according to sociodemographic characteristics, smoking history, and chronic disease.

Characteristics	N (%) (N = 201)	Emotional Exhaustion		Depersonalization		Personal Accomplishment	
		Mean (SE)	*p* Value *	Mean (SE)	*p* Value *	Mean (SE)	*p* Value *
Gender							
Male	70 (34.8%)	22.7 (1.4)	0.24	19.3 (1.4)	0.74	32.7 (1.3)	0.01
female	131 (65.2%)	24.8 (0.9)		18.1 (0.8)		28.9 (0.9)	
Age group							
20-	89 (44.3%)	28.3 (1.1)	<0.001	22.9 (1.0)	<0.001	27.1 (1.2)	<0.001
30-	73 (36.3%)	22.5 (1.3)		16.7 (1.2)		30.6 (1.1)	
40-	39 (19.4%)	17.5 (1.6)		1.6 (1.4)		36.7 (1.7)	
Residence							
Urban	180 (89.6%)	24.2 (0.8)	0.72	18.6 (0.8)	0.65	30.2 (0.8)	0.75
Rural	21 (10.4%)	23.1 (2.8)		17.3 (2.1)		31.2 (2.3)	
Marital Status							
Single	78 (38.8%)	26.6 (1.1)	0.01	22.8 (1.1)	<0.001	27.1 (1.1)	0.001
Married	123 (61.2%)	22.5 (1.1)		15.7 (0.9)		32.2 (0.9)	
Smoking History							
Non-smokers	187 (93%)	24.2 (0.8)	0.56	18.3 (0.8)	0.48	30.2 (0.8)	0.95
Smokers	14 (7%)	22.4 (2.9)		20.7 (3.1)		30.9 (2.5)	
Chronic disease							
Yes	31 (15.4%)	24.3 (2.2)	0.90	19.5 (2.2)	0.70	32.2 (2.1)	0.25
No	170 (84.6%)	24.1 (0.8)		18.3 (0.8)		29.8 (0.8)	

* Mann Whitney test.

**Table 3 ijerph-18-05368-t003:** Distribution of the doctors’ Maslach Burnout Inventory mean scores according to some workplace characteristics and COVID-19 case exposure.

Characteristics	*N* (%) (*N* = 201)	Emotional Exhaustion		Depersonalization		Personal Accomplishment	
		Mean (SE)	*p* Value *	Mean (SE)	*p* Value *	Mean (SE)	*p* Value *
Job title							
Resident doctor	67 (33.3%)	27.4 (1.3)	0.01	23.6 (1.3)	<0.001	26.1 (1.4)	<0.001
Assistant lecturer	74 (36.8%)	25.9 (1.2)		19.0 (1.1)		29.9 (1.2)	
Specialist/Lecturer	25 (12.4%)	18.7 (2.3)		12.5 (1.5)		34.5 (1.8)	
Assistant prof/Professors	35 (17.4%)	17.9 (1.6)		11.9 (1.4)		35.8 (1.8)	
Working specialty:							
Medical	92 (45.8%)	22.9 (1.1)	0.41	17.2 (0.9)	0.37	31.9 (1.1)	<0.001
Surgical/Anesthesia and ICU	59 (29.4%)	24.9 (1.6)		19.8 (1.6)		32.8 (1.5)	
Academic	50 (24.9%)	25.2 (1.5)		19.3 (1.5)		23.9 (1.5)	
Direct exposure with COVID-19 case							
Yes	63 (31.3%)	26.1 (1.4)	0.08	21.6 (1.4)	0.01	30.7 (1.3)	0.78
No		23.1 (0.9)		17.1 (0.8)		30.0 (0.9)	
Working system							
Morning Shifts	85 (42.3%)	22.4 (1.2)	0.06	16.5 (1.2)	0.02	31.3 (1.1)	0.24
Mixed shifts	115 (57.5%)	25.4 (1.1)		19.9 (0.9)		29.4 (1.1)	
Working hours per day							
<4 h	26 (12.9%)	15.8 (2.2)	<0.001	11.5 (1.8)	<0.001	34.6 (1.9)	0.04
4–8 h	95 (47.3%)	23.2 (1.1)		16.3 (0.9)		30.6 (1.1)	
>8 h	80 (39.8%)	27.8 (1.2)		23.3 (1.1)		28.3 (1.3)	

* Mann Whitney test.

## Data Availability

Data is available upon request for ethical purposes.

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
