# Peer review of "Burnout Impact of COVID-19 Pandemic on Health-Care Professionals at Assiut University Hospitals, 2020"

_ijerph, 2021, doi:10.3390/ijerph18105368_

Round 1

Reviewer 1 Report

This paper  is properly structured and is deals with a very important issue - the consequences of covid-19 among medical personel. The order of the steps of the research procedure is rather correct, the applied research methods  do not raise major objections. The authors have well recognized the limitations of this study. From a technical point of view the article is correct, interesting to read.

Critical remarks:

- The title of the publication is compliance with its content.  The research problem and objective are not formulated clearly and precisely - is it only a diagnosis of the current state?

- There is no information about the socio-cultural determinants of care for medical personnel in the field of BOS prevention. Are there any system solutions? Have there been any attempts to introduce such solutions?

- The key (last) part of the article does not contain recommendations / proposed solutions. Very synthetic conclusions.

- I recommend greater internationalization of literature sources and their greater variety.

Summary:

The originality of presented research results is poor, but the value of research for applied science and its practical applicability - when possible areas for implementation are identified - excellent.

Reviewer 2 Report

The topic is interesting and very current. Please look at these points:

  1.  Lines 136-137: "44.3 % of the enrolled doctors were aged 20 - 30 years, 36.3% aged 30- 40 years and 19.4 % aged more than 40 years". Only 201 doctors answered questionnaire, and only 19.4% were older than 40 y.o. Older doctors did not answered more than younger the questionnaire, this can be a limitations of the study and must be reported.
  2. Lines 81-87: "longer working hours shortage of PPE, heavy workload, extensive responsibilities, lack of specific drugs and protocols and care for unstable patients". Some doctors changed also their locations and tasks, as also chronic diseases were pushed aside. Please look at these 2 important papers:      Intracranial hemorrhage and COVID-19, but please do not forget "old diseases" and elective surgery. Brain Behav Immun. 2021 Feb;92:207-208. doi: 10.1016/j.bbi.2020.11.034.      The COVID-19 emergency does not rule out the diagnostic arsenal in intracerebral hemorrhage: Do not forget the old enemies. Brain Behav Immun. 2021 Jan;91:792-793. doi: 10.1016/j.bbi.2020.11.020.
  3. Lines 196-198: Figure legend Table 3 is so long and dispersive. Please improved.
  4. Conclusion is too short. Reported briefly your data. 
  5. Lines 245-247: "physicians in a tertiary hospital in Saudi Arabia concluded that the most affected physicians with the burnout syndrome were resident physicians" Please improved ref "The emotional impact of COVID-19: From medical staff to common people"
  6. Lines 284-285: "To our knowledge this is the first time to study this topic in our institution during this pandemic". This apports little to the literature, please remove.
  7. Lines 293-335: "Strengths and Limitations" This paragraph must be put before Conclusions section.

Reviewer 3 Report

Dear authors,

Your manuscript is interesting but I need you to answer some questions:

ABSTRACT

  • The abstract methodology is confusing. Authors should describe it better.

MATERIALS AND METHODS

Ethical considerations:

  • One clinical trial has been registered but the design is cross-sectional. The authors should clarify this.

DISCUSSION

  • The authors should better explain the differences between academic and healthcare personnel. Due to the difference in functions it is important to analyze it.
  • The authors have not explained how to apply the research to clinical practice.
  • The authors have not included research limitations.

REFERENCES

  • Some references that have errors. The authors should review this section.

Round 2

Reviewer 2 Report

Authors solved all my criticisms.

Reviewer 3 Report

Dear authors,

Thanks for your reply. The explanations of the authors are satisfactory. The paper has greatly improved its quality.

Congratulations on your work.

Best regards